# Clinical and Microbiological Efficacy of Pyrophosphate Containing Toothpaste: A Double-Blinded Placebo-Controlled Randomized Clinical Trial

**DOI:** 10.3390/microorganisms8111806

**Published:** 2020-11-17

**Authors:** Inpyo Hong, Hyun Gee Lee, Hye Lim Keum, Myong Ji Kim, Ui-Won Jung, KiJung Kim, Su Yeon Kim, Taehun Park, Hye-Jin Kim, Jin Ju Kim, Woo Jun Sul, Susun An, Jae-Kook Cha

**Affiliations:** 1Department of Periodontology, Research Institute for Periodontal Regeneration, Yonsei University College of Dentistry, 50 Yonsei-ro, Seodaemun-gu, Seoul 120-752, Korea; inpyo@yuhs.ac (I.H.); mjikim212@yuhs.ac (M.J.K.); drjew@yuhs.ac (U.-W.J.); 2Safety and Microbiology Lab, Amorepacific R&D Safety and Microbiology Lab, Amorepacific Corporation R&D Center, Yongin 17074, Korea; hglee@amorepacific.com; 3Department of Systems Biotechnology, Chung-Ang University, Anseong 17546, Korea; hyelim0904@gmail.com (H.L.K.); kijungkim@cau.ac.kr (K.K.); suyeonkim@amorepacific.com (S.Y.K.); taehunpark@amorepacific.com (T.P.); hjinkim327@gmail.com (H.-J.K.); winypuen9762@gmail.com (J.J.K.); sulwj@cau.ac.kr (W.J.S.)

**Keywords:** pyrophosphate, toothpaste, calculus, microbiome, dental plaque, dysbiosis

## Abstract

(1) Background: Dental calculus works as a niche wherein pathogenic bacteria proliferate in the oral cavity. Previous studies revealed the anticalculus activity of pyrophosphates, however there was no clinical study that evaluated microbiome changes associated with calculus inhibition. Therefore, the aim of this randomized clinical trial was to evaluate the calculus inhibition of pyrophosphate-containing toothpaste and its effect on oral microbiome changes. (2) Methods: Eighty subjects with a calculus index ≥2 on the lingual of the mandibular anterior tooth were randomly allocated to the test group that pyrophosphate-containing toothpaste was given to or the placebo control group. Full mouth debridement and standardized tooth brushing instruction were given before the allocation. Plaque index, gingival index, calculus index, probing depth, and bleeding on probing were measured at the baseline, and at 4, 8 and 12 weeks. Genomic DNA was extracted from the plaque samples collected at the baseline and at 12 weeks, and 16S ribosomal RNA gene amplicon sequencing was applied for microbiome analysis. (3) Results: None of the clinical parameters showed significant differences by visits or groups, except the plaque index of the test group, which reduced significantly between 4 and 12 weeks. A significant difference of microbiome between the baseline and 12 weeks was observed in the test group. Between baseline and 12 weeks, the proportion of *Spirochetes* decreased in the control group, and the proportions of *Proteobacteria*, *Fusobacteria* and *Spirochetes* in the phylum level and the proportions of *Haemophilus*, *Fusobacterium* and *Capnocytophaga* in the genus level decreased in the test group. In the test group, as plaque index decreased, *Streptococcus* increased, and *Fusobacterium* and *Haemophilus parainfluenza* decreased. (4) Conclusion: The use of pyrophosphate-containing toothpaste effectively inhibited the dysbiosis of the oral microbiome and the proliferation of pathogenic species in periodontal disease. Clinically, plaque formation in the pyrophosphate-containing toothpaste group was effectively decreased, however there was no significant change in calculus deposition.

## 1. Introduction

Periodontitis is the result of the host immune response against microbial challenge caused by oral microbiome dysbiosis [1]. It is now widely accepted that periodontal inflammation is due to the collapse of homeostasis in the healthy oral microbiome, not an infection caused by some specific pathogenic bacteria [2]. Through previous observational studies, it was found that pathogenic anaerobic species and the diversity of the oral microbiome are increased in periodontitis patients [3,4]. The increase in pathogenic microbial challenge caused by dysbiosis intensifies periodontal inflammation, and vice versa, periodontal tissue destruction caused by inflammation offers a favorable microenvironment for dysbiosis, with abundant nutrition from the tissue breakdown products [1].

Dental calculus is a crystallized product of calcium phosphate supersaturation, thus calculus itself has not been recognized as a pathogen of periodontal disease. However, since the surface of the calculus is rough, a plaque layer is always formed in the oral environment, and as a result, dental calculus acts as a niche of bacterial proliferation, providing a favorable environment for pathogenic dysbiosis [5]. For this reason, the aim of clinical treatment procedures for periodontal disease, such as scaling and root planning, is to remove and inhibit calculus deposition from the tooth surface [6].

Previous studies have reported that the application of pyrophosphate in self-performed plaque control can inhibit the formation of dental calculus [7]. Pyrophosphate interrupts the conversion of amorphous calcium phosphate to hydroxyapatite, which consists of dental calculus [8]. With the use of the pyrophosphate-containing dentifrice, a previous study reported that the formation of calculus was reduced by 40% after a year [9]. In addition to the reduced amount of calculus, it was confirmed by scanning electron microscope imaging that the structure of calculus was less compact when pyrophosphate was applied [10].

It is clear that the regular removal of dental calculus as a part of maintenance therapy helps in maintaining periodontal health [11]. Furthermore, various studies support the inference that periodontal inflammation interacts with systematic diseases, such as diabetes mellitus and cardiovascular disease [12,13]. For this reason, not only the removal of calculus but also the inhibiting of the deposition of calculus through proper supportive periodontal therapy are important factors in maintaining periodontal health, and its related systematic health [14].

The ultimate purpose of inhibiting dental calculus formation is to prevent pathogenic changes in the oral microbiome, which lead to periodontal disease. Thus, in addition to the evaluation of clinical calculus formation, the evaluation of pathogenic changes in the oral microbiome is necessary. To the best of our knowledge, there has been no clinical study that evaluates calculus inhibition and the associated microbiome changes. Therefore, the aim of the present study was to evaluate the calculus inhibition of pyrophosphate-containing toothpaste clinically and its effect on oral microbiome changes.

## 2. Materials and Methods

### 2.1. Study Design and Population

This study was designed as a double-blinded placebo-controlled randomized clinical trial. The standardization of clinical measurement between investigators was ensured via calibration meetings before the experiment. All procedures followed the Declaration of Helsinki and good clinical practice guidelines and were approved by the Institutional Review Board of the Yonsei University Dental Hospital (2-2017-0049). Eighty subjects were enrolled from the Department of Periodontology, Yonsei University Dental Hospital, from April 2018 to January 2019. Before enrollment, all participants were informed about the nature of the study, and an informed consent form was obtained. The CONSORT flowchart is presented in Figure 1. At the screening, clinical index measurement, plaque collection and full mouth debridement with ultrasonic scaler were performed before allocation.

### 2.2. Sample Size Estimation

The required sample size was calculated using G*Power 3.1 software [15]. A minimum sample size of 27 patients in each group was needed to detect a clinically relevant difference of calculus formation between the groups with a statistical power of 80% and a significant level of 5% (SD = 0.5) based on the previous study [16]. A dropout rate of 20% was estimated and the minimal required sample size was determined to be 68 patients. Regarding minimal required sample size, the total required sample size was determined to be 80 patients.

### 2.3. Randomization and Blinding

The participants were randomly allocated to one of two study groups, forty patients in each group, by a random numbering sequence generated with web-based software (www.sealedenvelope.com). A double-blind study was performed so that the examiners were not involved in the group assignment. Before the assignment, standardized tooth brushing instruction was given to all participants. Follow up visits were at 4 weeks (Visit 2), 8 weeks (Visit 3) and 12 weeks (Visit 4). Up until 12 weeks, all participants were required to use the given toothpaste according to the assignment only.
Test group: Toothpaste containing 3.4% tetrasodium pyrophosphate was given.Control group: Placebo toothpaste without pyrophosphate was given.

Except for tetrasodium pyrophosphate, the ingredients of the toothpaste in both groups were identical.

### 2.4. Inclusion and Exclusion Criteria

This study was performed on patients with visually confirmed dental calculus and who were diagnosed as having gingivitis or incipient periodontitis. The further inclusion criteria were as follows: (1) being >18 and <65 years of age and in good general health, (2) having a minimum of 18 teeth, (3) having calculus index ≥2 on the lingual surface of mandibular anterior teeth, and (4) being able to perform adequate tooth brushing. The exclusion criteria were as follows: (1) not providing written informed consent, (2) being pregnant or lactating, (3) having an oral mucosal disease such as lichen planus, (4) being diagnosed as chronic, moderate or advanced periodontitis, (5) smoking, (6) being allergic to the chemical ingredient of conventional toothpaste and (7) being judged as being unsuitable for study inclusion by the clinician for some other reasons.

### 2.5. Baseline Measurement and Follow up Measurements

Plaque collection was performed according to the previous study [17]. One hour before collection, oral hygienic control, such as mouth rinses or brushing and eating, was prevented. Before swabbing, participants rinsed their mouths with pure water. Dental plaque samples were collected by swabbing along the lingual side of the lower anterior teeth surfaces. The acquired swabs were immediately stored at −80 °C until being analyzed.

After plaque collection, a representative six Ramfjord teeth (#16, 21, 24, 36, 41, and 44) were examined for clinical index measurement. Plaque index (PI, Loe, 1967), gingival index (GI, Loe. 1967), calculus index (CI, Greene and Vermillion, 1964), probing depth (PD) and bleeding on probing (BOP) were scored for each Ramfjord tooth [18,19]. Since lingual side of tooth #41 clinically shows pronounced deposition of dental calculus due to inaccessibility of tooth brushing and proximity to sublingual saliva gland, CI of #41 lingual was analyzed separately in addition to total CI. After clinical index measurement, full mouth debridement and standardized tooth brushing instruction were performed, and patients were randomly allocated to the test or the control group.

At the 4-weeks and 8-weeks follow up visits, only clinical index measurement was performed. At 12 weeks, plaque collection and clinical index measurement were performed (Figure 2). 

### 2.6. Genomic DNA (gDNA) Extraction

Bacterial gDNA extraction was conducted according to the modified protocol of DNeast Blood and Tissue Kit (QIAGEN). Each plaque sample was transferred to a 1.5 mL screw-capped tube. Then, 400 µL digestion buffer (20 mM Tris-HCl, pH 8.0, 2 mM EDTA, and 1.2% Triton X-100) containing lysozyme (20 mg/mL) was added and incubated at 37 °C for 30 min. To each tube, 400 µL AL buffer and proteinase K (final conc. 1.23 mg/mL) were added and slightly vortexed for 5 s at each step. After incubation for 13–14 h overnight, incubation was performed for 5 min in a 95 °C heat-block for proteinase inactivation. Afterward, 445 µL of 100% ethanol was added per tube, and the samples were vigorously mixed by vortexing. The subsequent steps followed the purification of total DNA from animal tissues (Spin-Column Protocol) of the kit. Finally, the extracted gDNA was eluted with 30 µL of AE buffer per tube and stored at −20 °C until use.

### 2.7. Sample Preparation and Sequencing

PCR with primers 518F and 927R was used to amplify the V4–V5 region of the bacterial 16S rRNA gene. The primer sequences for 518F and 927R are (5′-CCAGCAGCYGCGGTAAN-3′) and (5′-CCGTCAATTCNTTTRAGT-3′), respectively. The thermal cycling conditions were as follows: initial denaturation (95 °C for 3 min), 33 cycles (95 °C for 30 s, 55 °C for 30 s, and 72 °C for 5 min), and final extension (72 °C for 5 min). Using AMPure XP beads (Beckman Coulter Ltd., High Wycombe, UK), the PCR products were purified. To barcode each sample, using the i7 and i5 index adapters of the Illumina Nextera XT Index Kit v2, indexing PCR was performed. Amplification cycles were performed only 8 times and the other thermal cycling conditions were the same as described above. Purifying the products was performed under the same conditions. Using the Illumina MiSeq platform, paired-end sequencing (2 × 300 bp) was performed by Macrogen Inc. (Seoul, Korea).

### 2.8. Microbiome Analysis and Statistical Anaylsis

Microbiome sequences were processed using the plugins in the QIIME^TM^ 2 (Quantitative Insights Into Microbial Ecology) pipeline 2019.7 [20]. Primer sequences were removed from the bacterial sequences with cutadapt 2.3 (Martin 2011 [21]) by using the default values. Reads that contained no bacterial primer sequence or a poor-quality primer sequence (i.e., error rate >10%) were eliminated at this step [21]. Paired-end sequence reads were denoised and merged at the truncated position of 222bp and 171bp for forward and reverse read sequences, respectively, via the dada2 denoise-paired process, whereby bacterial amplicon sequence variants (ASVs) were identified [22]. The feature-classifier classify-sklearn (4) was used to assign bacterial taxonomy via the Greengenes database at 99% sequence similarity (mitochondrial or chloroplastic ASVs were eliminated). Aligning of the ASVs was performed by using the phylogeny align-to-tree-mafft-fasttree. With a rarefied depth of 2164 reads per sample, α-diversity (Chao1; observed ASVs) and β-diversity were determined.

### 2.9. Statistical Anaylsis

For clinical parameters, statistical analysis was performed with commercially available software (IBM SPSS Statistics 23, SPSS, Chicago, IL, USA). The Mann–Whitney U test and chi-square test were applied to compare demographic characteristic and clinical index between the control and test groups. To assess the significant difference in the bacterial community (β-diversity) between the groups, analysis of similarity (ANOSIM) was used. To determine the significantly different alpha diversities and taxonomies between the groups, the Wilcoxon’s rank-sum test or the *t*-test was performed. To identify the significant ASVs in each group, linear discriminant analysis (LDA) and effect size (LEfSe) analysis was carried out with an LDA score ≥2.5. Linear regression analysis of specific strains and PI was performed. Since buccal side of teeth clinically show lessened deposition of dental plaque due to easiness of oral hygiene control, the regression analysis was done separately for buccal and lingual PI.

## 3. Results

A total of 80 participants were enrolled in the study. Five participants in the control group and one in the test group were excluded from the analysis due to missing follow-up visits. In consequence, the study was completed with 74 participants (*n* = 39 and 35 in the test and control groups, respectively). The demographic characteristics of the participants are presented in Table 1. There was no significant difference in age and distribution of gender. 

The clinical parameters are presented in Table 2. None of the clinical parameters differed significantly between the test and control groups at the baseline and each follow-up visit. Changes in the clinical parameters are depicted in Figure 3. Since full mouth debridement was done at the baseline, all clinical parameters were reduced between the baseline and 4 weeks. Reduced clinical parameters were maintained until 12 weeks in both the test and control groups. The PI of the test group was reduced significantly between 4 weeks (2.77) and 12 weeks (1.86), while there was no statistically significant change in the PI of the control group. Except for PI, there were no significant changes in clinical parameters between 4 weeks and 12 weeks in either the test or the control group. Also, there was no significant difference between clinical parameters of the test and the control group in all visits.

At baseline, the microbiome of the test and control groups were similar (Figure 4A). In the control group, there was no significant difference in microbiome between baseline and 12 weeks (ANOSIM, *p* = 0.689; Figure 4B Left). In contrast, a significant difference in microbiome between the baseline and 12 weeks was observed in the test group (ANOSIM, *p* = 0.001; Figure 4B, Right). The alpha diversity of the control and the test group was reduced between baseline and 12 weeks, but there was no statistical significance (Figure 4C).

In detail, a total of 21 phyla and 205 genera was identified in the total plaque samples. The phylum-level taxonomic analysis of the test group and the control group is summarized in Table 3. At the phylum level, *Firmicutes* and *Proteobacteria* were dominant in both the test and the control group. In the control group, the proportion of *Spirochaetes* decreased significantly between the baseline and 12 weeks. In the test group, the proportions of *Proteobacteria*, *Fusobacteria* and *Spirochetes* decreased significantly between the baseline and 12 weeks. In contrast, the proportion of *Firmicutes* in the test group increased significantly between the baseline and 12 weeks.

The genus-level taxonomic analysis of the test group and the control group is summarized in Table 4. At the genus level, *Streptococcus* was dominant in both the test and the control group. In the control group, there was no significant change in genus distribution. In the test group, the proportion of *Streptococcus* was significantly increased between baseline and 12 weeks. The proportions of *Haemophilus, Fusobacterium* and *Capnocytophaga* were significantly decreased between baseline and 12 weeks.

Linear discriminant analysis effect sized (LEfSe) was applied for amplicon sequence variants (ASVs) level analysis. In total, 29 ASVs were more abundant at the baseline, and 20 ASVs were more abundant at 12 weeks. The LEfSe results are summarized in Appendix A. The species that show significant changes in LEfSe between baseline and 12 weeks are summarized in Appendix A. In the control group, a small number of species showed significant change between baseline and 12 weeks. In the test group, in contrast to the control group, a large number of species showed significant change between baseline and 12 weeks. Considering the result of the taxonomic analysis, *Haemophilus parainfluenzae* and *Fusobacterium* were significantly decreased between baseline and 12 weeks, while *Streptococcus* was significantly increased in the test group.

Linear regression analysis of specific strains and PI values was performed and is presented in Figure 5. PI values that showed a significant change were selected as a clinical indicator, and the taxa that showed a significant change in LEfSe between baseline and 12 weeks were selected. In the control group, only in buccal, it was confirmed that the PI value decreased as *Fusobacterium* decreased at 12 weeks as compared to baseline, and decreased as the *Streptococcus* increased. In the buccal of the test group, the PI value decreased as *Haemophilus parainfluenzae* decreased at 12 weeks compared to baseline. On the contrary, it was confirmed that the PI value decreased as *Streptococcus* increased. In the case of lingual, it was confirmed that the PI value decreased as *Fusobacterium* decreased at 12 weeks compared to the baseline. On the contrary, the PI value decreased as *Streptococcus* increased.

## 4. Discussion

Through this study, it was confirmed that the use of pyrophosphate-containing toothpaste effectively inhibited the dysbiosis of the oral microbiome and the proliferation of pathogenic species such as *fusobacterium* and *capnocytophaga*. Clinically, the plaque formation of pyrophosphate-containing toothpaste group was effectively decreased, however there was no significant change in calculus deposition. The use of pyrophosphate-containing toothpaste demonstrated its efficacy in maintaining a healthy oral microbiome, which is related to the prevention of periodontal disease.

Pyrophosphate inhibits dental calculus formation by interrupting the conversion of amorphous calcium phosphate to hydroxyapatite crystal [8]. Since calculus has a rough surface, a new bacterial plaque attaches to the generated calculus [5]. Plaques absorb calcium ions and phosphates in oral fluid to be calcified and changed to hard calculus. Calculus offers a favorable microenvironment for the proliferation of pathogenic species, which are mostly anaerobic, so they cannot survive without polymicrobial synergy and dysbiosis [2].

In the present study, the use of pyrophosphate-containing toothpaste did not result in a significant difference in CI. Since CI is non-continuous and has ordinal values of integers from 0 to 3, it is difficult to express an intermediate state between the criteria of the scores and the quality of calculus. Thus, only a dramatic change in calculus deposition can be reflected in the change of CI. Since various factors, such as the composition and secretion rate of saliva, and diets, effect dental calculus formation, there was a limitation in showing significant changes in calculus formation with pyrophosphate-containing toothpaste [5].

Unlike CI, a significant difference in PI was observed. When calculus is produced in the oral cavity, the surface always remains covered by non-calcified dental plaque [6]. The inhibition of crystallization by pyrophosphate results in a less compact surface of the deposited calculus [10]. This phenomenon makes it difficult for plaque to accumulate on the surface of calculus, and the attachment between plaque and calculus is weakened. Weak plaque–calculus attachment allows easy plaque removal with the same tooth brushing method. With this mechanism, the use of pyrophosphate-containing toothpaste not only inhibits plaque deposition, but also facilitates the removal of plaque.

In the analysis of phylum level, the use of pyrophosphate-containing toothpaste decreased the portion of *Spirochaetes* in the oral microbiome, while the control group showed an increase of *Spirochaetes*. According to previous studies comparing the bacterial profiles of periodontitis and healthy patients, *Spirochaetes* were highly associated with the diseased state [4]. *Spirochaetes* are small, have motility, and inhibit the polymorphonuclear leukocyte (PMNL) invasion of the host, so they can easily invade the periodontal tissue and avoid host immune surveillance [23]. Among the *Spirochaetes*, the *Treponema* genera, such as *Treponema denticola* and *Treponema vincentti*, are among the more highly documented periodontal pathogenic species [24]. In particular, *Treponema denticola* is a representative periodontal pathogenic species that is classified as ‘red complex’ with *Porphyromonas gingivalis* and *Tannerella forsythia* [25]. Red complex species induce tissue destructive enzymes and immune responses, and products from tissue destruction create acidic and anaerobic conditions. Since most pathogenic species are anaerobic, the anaerobic condition induced by tissue destruction is hospitable for the dysbiotic and virulent change of the microbiome [1].

In the analysis of the genus level, the proportion of *Streptococcus* was significantly increased with the use of pyrophosphate-containing toothpaste. *Streptococcus* is highly associated with a healthy state, and suppresses the increase in ‘red complex’ through antagonistic action [4]. Since *Porphyromonas gingivalis* cannot maintain its growth below pH 6.5, the acidic environment in which *Streptococcus mutans* forms in its metabolism inhibits the pathogenic change related to *Porphyromonas gingivalis* [26]. Furthermore, a previous in vitro study showed that the presence of *Streptococcus* inhibits the growth of *Tannerella forsythia* and *Prevotella intermedia* [27]. With these mechanisms, *Streptococcus* protects the homeostasis of the microbiome from dysbiosis in the direction of increasing pathogenic species [28]. In the aspect of dental caries, *Streptococcus mutans* is well-known as a key cariogenic species, so that there could be a limitation to simply regarding *Streptococcus mutans* as a protector of a healthy microbiome [29]. However, a recent study based on DNA sequencing failed to find a difference in *Streptococcus mutans* proportion between caries-active and caries-free subjects [30].

Pyrophosphate-containing toothpaste reduced *Fusobacterium* and *Capnocytophaga*. *Fusobacterium* can co-adhere with most of the species, and is an intermediate between the early colonizer and late colonizer [31]. Once the pellicles are formed, the early colonizer, mostly *Streptococcus*, is the first to attach. *Streptococcus* is antagonistic to ‘red complex’, but *Fusobacterium* acts as an intermediary, allowing the attachment of ‘red complex’. In addition to assisting pathogenic bacteria in attaching, *Fusobacterium* itself upregulates the expression of tissue destructive enzymes, such as matrix metalloproteinase, and secretes serine protease, causing tissue destruction [32]. *Capnocytophaga* degrades IgA and IgG, and induces the alteration of PMNL function [33]. Previous studies found a significant increase in *Capnocytophaga* in various forms of periodontitis, compared to the healthy state [34,35].

Linear regression analysis confirmed that the PI value and the specific bacteria were closely related. *Haemophilus parainfluenzae* and *Fusobacterium* were negatively related to PI, while streptococcus was positively related to PI. *Haemophilus parainfluenzae* is known as an opportunistic pathogen and is related to the progression of the periodontitis [36] Furthermore, as previously discussed, *Streptococcus* and *Fusobacterium* are in an antagonistic relation. However, there is a limitation to interpretation, as plaque accumulation and changes of such species are in a causal relationship. From our study, we could regard *Streptococcus mutans* as a beneficial indicator, and *Haemophilus parainfluenzae* and *Fusobacterium* as harmful indicators. However, further pathological investigation is needed to prove the causal relationship between such species and plaque-induced gingivitis.

Besides the observational study of bacteria, *Hsu* et al. reported that pyrophosphate suppresses bacterial growth by the inhibition of F-ATPase and chelation of Mg^2+^ [37]. F-ATPase transports protons out of cells, so it is related to the acid tolerance of bacteria [38]. The homeostasis of Mg^2+^ stabilizes membrane and macromolecular complexes. Since Gram-negative and Gram-positive have different regulators for low extracytoplasmic Mg^2+^, the chelation of Mg^2+^ has different effects. In Gram-negative species, low Mg^2+^ promotes the transcription of several genes involved in virulence and adaptation. In Gram-positive species, sensor kinase is inactivated by low Mg^2+^, leading to the inactivation of virulence-associated genes [39]. Therefore, the inhibition of F-ATPase and low concentrations of Mg^2+^ due to chelation are deemed to induce the alteration of the composition of the microbiome, rather than just their bactericidal effect.

To the best of our knowledge, this study is the first randomized clinical trial to analyze the effect of pyrophosphate on the oral microbiome through 16S ribosomal RNA gene amplicon sequencing. The culturing method, which most of the previous studies used, has the limitation that it cannot replicate the complex oral microbiome because only one species can be cultured in the plate. Furthermore, there is the limitation that most of the pathogenic species are uncultivable. Using 16S ribosomal RNA gene amplicon sequencing offers comprehensive and accurate information about the oral microbiome, which interacts actively between species and the surrounding conditions in the oral environment.

## 5. Conclusions

In conclusion, the use of pyrophosphate-containing toothpaste effectively inhibited plaque accumulation, however, there was no significant inhibition in calculus deposition. Microbiologically pyrophosphate-containing toothpaste decreased pathogenic species in the oral microbiome. Anti-mineralization effect of pyrophosphate may count for plaque inhibition and preventing pathogenic dysbiosis. The use of pyrophosphate-containing toothpaste can be adequate for maintaining a periodontally healthy state.

## Figures and Tables

**Figure 1 microorganisms-08-01806-f001:**
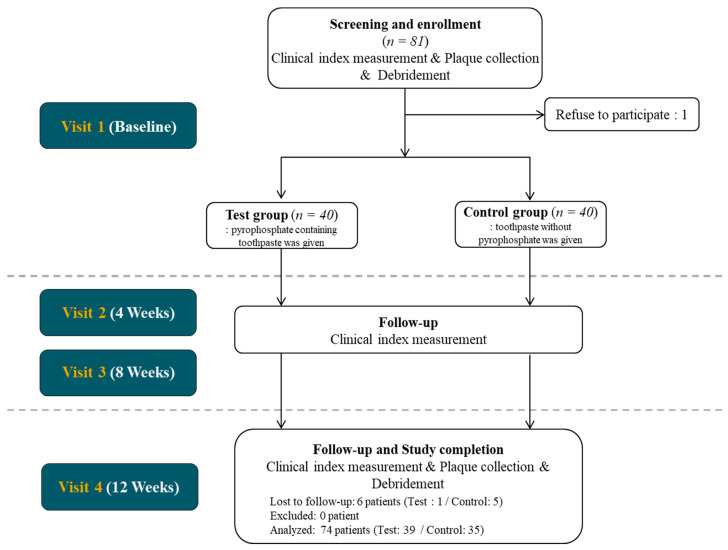
CONSORT Flowchart of the study.

**Figure 2 microorganisms-08-01806-f002:**
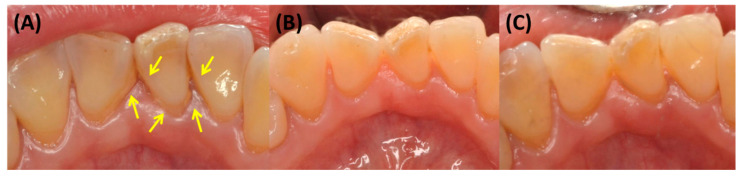
Clinical photo of #41 lingual at baseline (**A**), 4 weeks (**B**) and 12 weeks (**C**) of control group. Since full mouth debridement was done at baseline, calculus observed at 4 weeks and 12 weeks are newly deposited calculus after debridement. Deposited calculus are pointed out with yellow arrows in (**A**).

**Figure 3 microorganisms-08-01806-f003:**
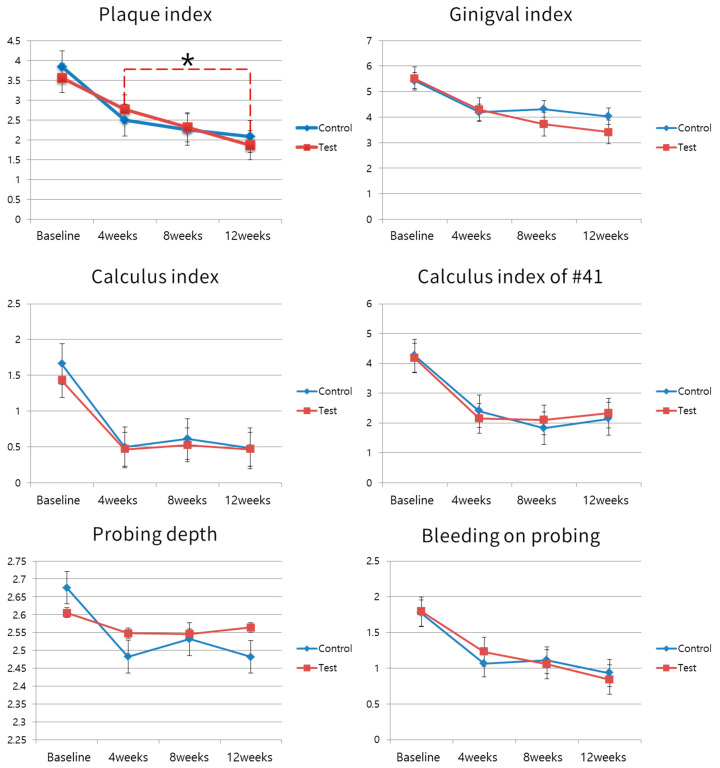
Changes of clinical parameters. Since full mouth debridement was given at a baseline visit, all clinical parameters were reduced significantly between baseline and 4 weeks. At the follow-up visits, all parameters showed no difference between test and control group. Only the plaque index of the test group between 4 weeks and 12 weeks showed a significant difference (* *p* < 0.05), while the other clinical parameters in each group did not show a significant difference among 4, 8 and 12 weeks of follow-up. The data presented in the graph for plaque index, gingival index, calculus index, and bleeding on probing are average per tooth, while those of probing depth are average per site.

**Figure 4 microorganisms-08-01806-f004:**
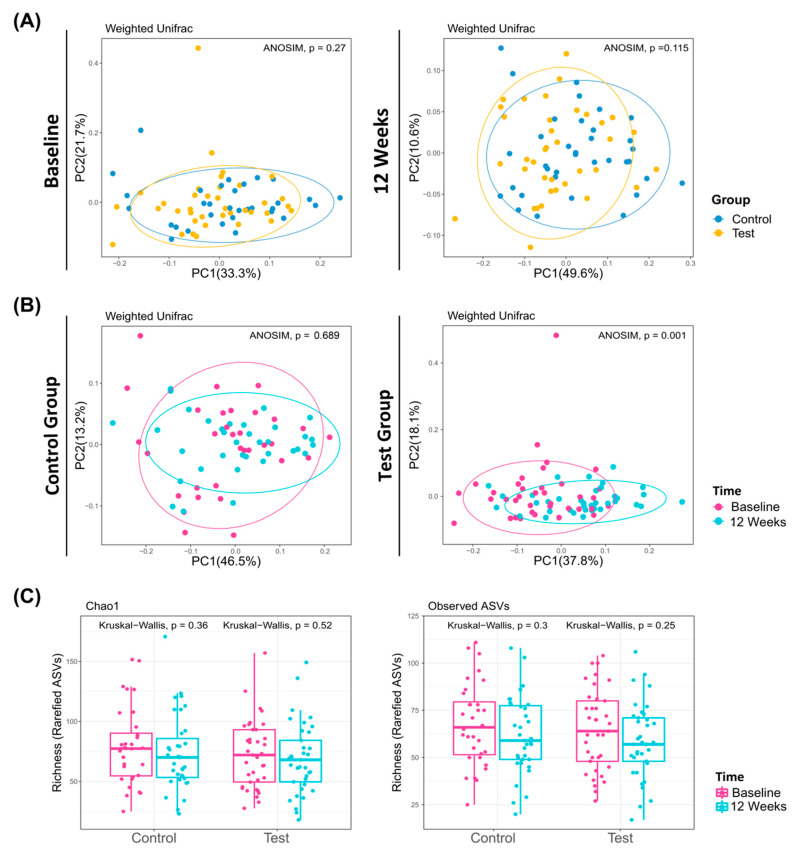
Evolution and richness of dental plaque microbiome. Principal coordinate analysis of microbiomes according to sampling time (**A**) and groups (**B**). (**A**) At baseline and 12 weeks, there was no significant difference between the control and test group. (**B**) In the test group, there was a significant difference between baseline and 12 weeks (ANOSIM, *p* = 0.001 < 0.05). (**C**) α-diversity of the control and test group at baseline and 12 weeks.

**Figure 5 microorganisms-08-01806-f005:**
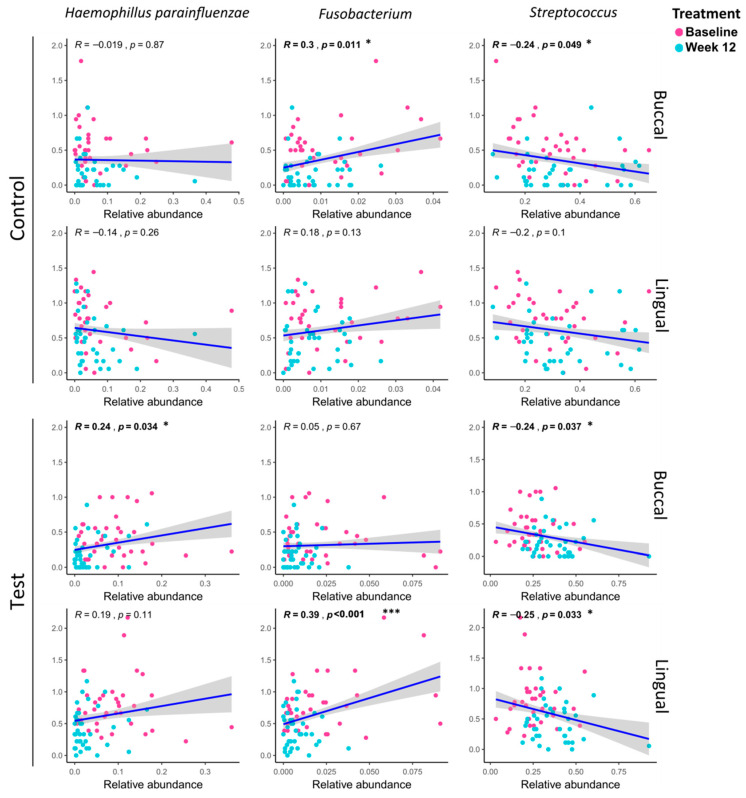
Linear regression analysis between plaque index (PI) and specific strains. The gray band surrounding the regression line shows the 80% confidence interval (80% CI) for the prediction of each data point in the horizontal axis. * Statistically significant correlation between plaque index and the specific strain (*p* < 0.05); *** Statistically significant correlation between plaque index and the specific strain (*p* < 0.001).

**Table 1 microorganisms-08-01806-t001:** Demographic character of participants.

	Control (*n* = 35)	Test (*n* = 39)	*p* Value (Control vs. Test)
Male/Female	14/21	19/20	0.14
Age	37.08 ± 11.08	33.72 ± 11.74	0.49

Ages of participants are presented as mean ± standard deviation. (*p* value = 0.14 > 0.05 and 0.49 > 0.05 in age and distribution of gender, respectively)

**Table 2 microorganisms-08-01806-t002:** Clinical parameters of test and control group.

		Baseline (before Debridement)	4 Weeks	8 Weeks	12 Weeks	*p* Value (4 Weeks vs. 12 Weeks)
PI	Control	3.84 ± 1.90	2.50 ± 1.68	2.26 ± 1.61	2.09 ± 1.52	0.33
	Test	3.56 ± 1.82	2.77 ± 2.07	2.32 ± 1.97	1.86 ± 1.46 *	0.037
	*p* value (Test vs. Control)	0.35	0.74	0.87	0.52	
CI	Control	1.66 ± 1.29	0.50 ± 0.54	0.61 ± 0.84	0.48 ± 0.38	0.58
	Test	1.43 ± 1.25	0.47 ± 0.50	0.53 ± 0.60	0.47 ± 0.42	0.73
	*p* value (Test vs. Control)	0.26	0.90	0.78	0.70	
CI 41	Control	4.26 ± 1.56	2.40 ± 2.44	1.83 ± 1.92	2.14 ± 1.65	0.92
	Test	4.18 ± 2.15	2.15 ± 2.29	2.10 ±2.26	2.33 ± 2.20	0.63
	*p* value (Test vs. Control)	0.66	0.75	0.84	0.98	
GI	Control	5.43 ± 2.54	4.19 ± 2.34	4.32 ± 2.42	4.03 ± 2.73	0.92
	Test	5.51 ± 2.98	4.30 ± 2.73	3.73 ± 2.64	3.42 ± 2.45	0.14
	*p* value (Test vs. Control)	0.83	0.78	0.42	0.28	
PD	Control	2.68 ± 0.30	2.48 ± 0.29	2.53 ± 0.25	2.48 ± 0.27	0.93
	Test	2.61 ± 0.32	2.55 ±0.29	2.55 ± 0.27	2.56 ± 0.21	0.67
	*p* value (Test vs. Control)	0.37	0.47	1.00	0.27	
BOP	Control	1.77 ± 1.16	1.07 ± 0.83	1.11 ± 0.89	0.93 ± 0.88	0.40
	Test	1.79 ± 1.40	1.23 ± 0.98	1.06 ± 0.98	0.84 ± 0.81	0.08
	*p* value (Test vs. Control)	0.93	0.53	0.51	0.56	

Clinical parameters are presented as mean ± SD. * Significant difference from 4 weeks. PI, average of plaque index per tooth; CI, average of calculus index per tooth; CI 41, average of calculus index of #41 tooth lingual; GI, average of gingival index per tooth; PD, average of probing depth per site; BOP, average of bleeding on probing per tooth.

**Table 3 microorganisms-08-01806-t003:** Phylum-level taxonomic analysis.

	Baseline			12 Weeks		
	Control (%)	Test (%)	Total (%)	Control (%)	Test (%)	Total (%)
*Firmicutes*	33.43	29.03	31.14	37.00	41.23 ***	39.17
*Proteobacteria*	37.8	40.04	38.97	33.63	28.45 ***	30.97
*Actinobacteria*	18.17	18.57	18.38	20.70	22.96	21.86
*Bacteroidetes*	7.94	8.67	8.32	6.71	5.81	6.25
*Fusobacteria*	2.48	2.93	2.71	1.84	1.41 **	1.62
*Spirochaetes*	0.11	0.55	0.34	0.01 **	0.03 *	0.02
*TM7*	0.02	0.03	0.03	0.07 ***	0.07	0.07
Others	0.05	0.17	0.11	0.04	0.04	0.04

Average proportions (%) of each phyla in total microbiome are presented. * Statistically significant difference between baseline and 12 weeks (*p* < 0.05); ** Statistically significant difference between baseline and 12 weeks (*p* < 0.01); *** Statistically significant difference between baseline and 12 weeks (*p* < 0.001).

**Table 4 microorganisms-08-01806-t004:** Genus-level taxonomic analysis.

	Baseline			12 Weeks		
	Control (%)	Test (%)	Total (%)	Control (%)	Test (%)	Total (%)
*Streptococcus*	30.00	25.65	27.73	33.15	31.69 **	35.48
*Lautropia*	12.37	12.63	12.51	8.18	10.62	8.81
*Actinomyces*	7.61	7.59	7.60	8.99	8.69	9.74
*Haemophilus*	7.37	9.42	8.44	6.08	6.44 **	4.53
*f_Neisseriaceae*	6.92	6.77	6.84	6.70	6.38	5.94
*Neisseria*	5.41	5.67	5.54	7.25	5.98	6.41
*Rothia*	4.8	4.93	4.87	4.84	5.23	5.57
*Corynebacterium*	4.55	4.98	4.77	4.46	4.65	4.53
*Capnocytophaga*	3.97	3.73	3.85	3.38	3.37 *	2.91
*f_Propionibacteriaceae*	1.08	0.95	1.01	2.20	1.42	1.82
*Kingella*	1.35	1.45	1.40	1.27	1.37	1.33
*f_Weeksellaceae*	1.38	1.09	1.23	1.45	1.26	1.28
*Fusobacterium*	1.22	2.21	1.73	0.77	1.25 **	0.79
Others	11.98	12.93	12.48	11.28	11.65	10.87

Average proportions (%) of each genus in total microbiome are presented. * Statistically significant difference between baseline and 12 weeks (*p* < 0.05); ** Statistically significant difference between baseline and 12 weeks (*p* < 0.001).

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
