# Peer review of "Clinical and Microbiological Efficacy of Pyrophosphate Containing Toothpaste: A Double-Blinded Placebo-Controlled Randomized Clinical Trial"

_microorganisms, 2020, doi:10.3390/microorganisms8111806_

Round 1
Reviewer 1 Report
|
The manuscript topic is actual and the paper has merit. It could be attractive, adequate and interesting for the journal readers. Please add at least 1 or 2 high quality clinical pictures like sample of before and post treatment image. It should be useful to have image related to each group in order to have a more attractive paper. DOI: 10.4103/0972-124X.94602 DOI: 10.7126/cdj.2013.1974
|
Author Response
The manuscript topic is actual and the paper has merit. It could be attractive, adequate and interesting for the journal readers.
Authors made excellent job realizing such interesting paper about Clinical and microbiological efficacy of pyrophosphate containing toothpaste trends and outcome, and absolutely the value of a RCT is high. However there are some minor concerns that authors should clarify in order to have a final more complete paper.
Clinical rationale of the paper should be highlighted at the beginning of the intro section, as well as the long term follow up of the treatment.
Introduction section should highlights the clinical rationale of this paper. Otherwise the study seems to be directed to just scientist or researcher and not to clinicians. Introduction section is poor. Add some more references about the periodontal health topic just published or actual.
Ans) Thank you for your kind review. We revised the paragraph in introduction section with adding new references as the reviewer recommended.
"It is clear that regular removal of dental calculus as a part of maintenance therapy helps maintaining periodontal health [11]. Also, various studies support that periodontal inflammation interact with systematic disease such as diabetes mellitus and cardiovascular disease [12,13]. In this reason, not only the removal of calculus but also inhibiting the deposition of calculus through proper supportive periodontal therapy is also an important factor in maintaining periodontal health and its related systematic health [14]. "
- Cobb, C.M. Clinical significance of non-surgical periodontal therapy: an evidence-based perspective of scaling and root planing. J Clin Periodontol 2002, 29 Suppl 2, 6-16, doi:10.1034/j.1600-051X.29.s2.4.x.
- Sanz, M.; Marco Del Castillo, A.; Jepsen, S.; Gonzalez-Juanatey, J.R.; D'Aiuto, F.; Bouchard, P.; Chapple, I.; Dietrich, T.; Gotsman, I.; Graziani, F., et al. Periodontitis and cardiovascular diseases: Consensus report. J Clin Periodontol 2020, 47, 268-288, doi:10.1111/jcpe.13189.
- Genco, R.J.; Borgnakke, W.S. Diabetes as a potential risk for periodontitis: association studies. Periodontol 2000 2020, 83, 40-45, doi:10.1111/prd.12270.
- Wang, Y.; Liu, H.N.; Zhen, Z.; Pelekos, G.; Wu, M.Z.; Chen, Y.; Tonetti, M.; Tse, H.F.; Yiu, K.H.; Jin, L. A randomized controlled trial of the effects of non-surgical periodontal therapy on cardiac function assessed by echocardiography in type 2 diabetic patients. J Clin Periodontol 2020, 47, 726-736, doi:10.1111/jcpe.13291.
Please add at least 1 or 2 high quality clinical pictures like sample of before and post treatment image. It should be useful to have image related to each group in order to have a more attractive paper.
Ans) As the reviewer suggested, we added clinical pictures in figure 2 to show changes of calculus deposition.
Thank you for tour kind review.
Reviewer 2 Report
I want to commend the authors on an interesting, readable and well-designed study. The manuscript is publishable with just a few revisions, Most of those are related to errors in the consistency and clarity of how data is presented.
Spacing around data is not consistent throughout the manuscript and on several of the figures and tables. This should be standardized at the very least with preference being a space both in front and behind of every = sign, ± and > sign. Also there should be a space between integers and units (e.g. 10 %).
Data should only be presented to the last significant figure and should be presented consistently for each unit or data type. For example; some p values include as few as 1 decimal place or as many as 5 decimal places. Surely including data to the 10,000th degree (0.0000X) is not needed. The appropriate standard should be used for all presented values for data of the same type. For example if all p values are presented with three decimal places then values beyond that should be listed as < 0.001.
Figures and tables should all indicate if data presented are sum, mean, median or another derived value. This is missing especially from the figures.
Captions should be more informative to thoroughly explain the figures. For example figure 5 doesn’t explain what shaded areas indicate nor the * values. This is especially a challenge with Figure 5 as there is not even an in text reference to that figure. The text should have references to all figures and captions should allow tables of figures to clearly be understood on their own.
Figure 2 should include bars that show the data range at each time point (like standard deviation). Single point values can be deceptive, and are no longer recommended for scientific publications.
Figure 4 is practically illegible and should be presented in another way or put in supplemental material in a expanded (and readable) format.
Remove “it was confirmed that” from line 270
Ginigivalis should be lowercase in line 313
Author Response
I want to commend the authors on an interesting, readable and well-designed study. The manuscript is publishable with just a few revisions, Most of those are related to errors in the consistency and clarity of how data is presented.
Spacing around data is not consistent throughout the manuscript and on several of the figures and tables. This should be standardized at the very least with preference being a space both in front and behind of every = sign, ± and > sign. Also there should be a space between integers and units (e.g. 10 %).
Ans) Thank you for your kind advice. We thoroughly reviewed the data presented on whole manuscript, and revised spaces as the reviewer recommended.
Data should only be presented to the last significant figure and should be presented consistently for each unit or data type. For example; some p values include as few as 1 decimal place or as many as 5 decimal places. Surely including data to the 10,000th degree (0.0000X) is not needed. The appropriate standard should be used for all presented values for data of the same type. For example if all p values are presented with three decimal places then values beyond that should be listed as < 0.001.
Ans) As reviewer suggested, we arranged our data to be consistent in decimal, and removed some unnecessary values presented. Generally, we rearranged our data in 100th degree which was thought to be enough to present our data.
Also we changed in Figure 5, which we used “p = 0.000061”, to “p < 0.001”, as reviewer pointed.
Figures and tables should all indicate if data presented are sum, mean, median or another derived value. This is missing especially from the figures.
Ans) As reviewer pointed, we revised captions and figure legends to indicate how data presented.
In table 1, we added a sentence in caption “Age of participants were presented as mean ± standard deviation.”
In table 2, we added a sentence in caption “Means of clinical parameters were presented” and clarified whether average of the clinical parameter is per tooth or per site.
In table 3 and 4, we added sentence in caption. “Average proportions (%) of each phyla (genus) in total microbiome were presented.”
Captions should be more informative to thoroughly explain the figures. For example figure 5 doesn’t explain what shaded areas indicate nor the * values. This is especially a challenge with Figure 5 as there is not even an in text reference to that figure. The text should have references to all figures and captions should allow tables of figures to clearly be understood on their own.
Ans) As reviewer pointed, we added explanation about figure 5 in caption, and clarified in paragraph which are about figure 5.
We added the sentence “Linear regression analysis of specific strains and clinical indicator values was performed and presented in Figure 5.” in manuscript.
Also we explained more in captions
“Linear regression analysis according to plaque index (PI) in control and test group.”
-> “Linear regression analysis according to a specific strain and plaque index (PI) in control and test group. The gray band surrounding the regression line shows the 95% confidence interval (95% CI) for prediction of each data point in the horizontal axis. * Statistically significant correlation between plaque index and the specific strain (p < 0.05); *** Statistically significant correlation between plaque index and the specific strain (p < 0.001)”
Figure 2 should include bars that show the data range at each time point (like standard deviation). Single point values can be deceptive, and are no longer recommended for scientific publications.
: As reviewer pointed, we changed our graph including bars of standard deviation. Since it is unavailable to upload figure in here, please check manuscript file.
Figure 4 is practically illegible and should be presented in another way or put in supplemental material in a expanded (and readable) format.
: As reviewer recommended, we decided to put figure 4 in supplements.
Remove “it was confirmed that” from line 270, Ginigivalis should be lowercase in line 313
: Revised as reviewer suggested. Thank you
Round 2
Reviewer 1 Report
Authors made excellent job addressing all the reviewers notes and request